# Development and validation of a cardiovascular risk prediction model for Sri Lankans using machine learning

**Chamila Mettananda[1]\*, Isuru Sanjeewa[2], Tinul Benthota Arachchi[2], Avishka Wijesooriya[2], Chiranjaya Chandrasena[2], Tolani Weerasinghe[2], Maheeka Solangaarachchige[3], Achila Ranasinghe[1], Isuru Elpitiya[1], Rashmi Sammandapperuma[1], Sujeewani Kurukulasooriya[1], Udaya Ranawaka[4], Arunasalam Pathmeswaran[5], Anuradhini Kasturiratne[5], Nei Kato[6], Rajitha Wickramasinghe[5], Prasanna Haddela[2], Janaka de Silva[4]**

1 Department of Pharmacology, Faculty of Medicine, University of Kelaniya, Ragama, Sri Lanka, 2 Faculty of Computing, Sri Lanka Institute of Information Technology, Malabe, Sri Lanka, 3 Computer Centre, Faculty of Medicine, University of Kelaniya, Ragama, Sri Lanka, 4 Department of Medicine, Faculty of Medicine, University of Kelaniya, Ragama, Sri Lanka, 5 Department of Public Health, Faculty of Medicine, University of Kelaniya, Ragama, Sri Lanka, 6 National Centre for Global Health and Medicine, Toyama, Shinjuku-ku, Tokyo, Japan

\* chamila@kln.ac.lk, chamilametta@hotmail.com

**Data Availability Statement:** The datasets used and analysed during the current study are available from the corresponding author or the Ethics

## Abstract

### Introduction and objectives

Sri Lankans do not have a specific cardiovascular (CV) risk prediction model and therefore, World Health Organization(WHO) risk charts developed for the Southeast Asia Region are being used. We aimed to develop a CV risk prediction model specific for Sri Lankans using machine learning (ML) of data of a population-based, randomly selected cohort of Sri Lankans followed up for 10 years and to validate it in an external cohort.

### Material and methods

The cohort consisted of 2596 individuals between 40–65 years of age in 2007, who were followed up for 10 years. Of them, 179 developed hard CV diseases (CVD) by 2017. We developed three CV risk prediction models named model 1, 2 and 3 using ML. We compared predictive performances between models and the WHO risk charts using receiver operating characteristic curves (ROC). The most predictive and practical model for use in primary care, model 3 was named "SLCVD score" which used age, sex, smoking status, systolic blood pressure, history of diabetes, and total cholesterol level in the calculation. We developed an online platform to calculate the SLCVD score. Predictions of SLCVD score were validated in an external hospital-based cohort.

### Results

Model 1, 2, SLCVD score and the WHO risk charts predicted 173, 162, 169 and 10 of 179 observed events and the area under the ROC (AUC) were 0.98, 0.98, 0.98 and 0.52

Review Committee of the Faculty of Medicine,
University of Kelaniya, Sri Lanka, Telephone no:
0112961267,email: ercmed@kln.ac.lk

**Funding:** This study was supported by the
Strengthening Research Outputs Grant of the
University of Kelaniya, Sri Lanka (RC/SROG/2021/
01). The funding bodies played no role in the
design of the study, collection, analysis, and
interpretation of data or in writing the manuscript.

**Competing interests:** The authors have declared
that no competing interests exist.

respectively. During external validation, the SLCVD score and WHO risk charts predicted
56 and 18 respectively of 119 total events and AUCs were 0.64 and 0.54 respectively.

## Conclusions

SLCVD score is the first and only CV risk prediction model specific for Sri Lankans. It predicts the 10-year risk of developing a hard CVD in Sri Lankans. SLCVD score was more
effective in predicting Sri Lankans at high CV risk than WHO risk charts.

## Introduction

There are no cardiovascular (CV) risk prediction models specific to or derived from Sri Lankans. Therefore, different risk prediction models derived from white Caucasians, or models
developed for the Southeast Asia region (SEAR), are being used for CV risk stratification of Sri
Lankans, which is not ideal. It has been shown that the risk stratification of the same Sri Lankan cohort using World Health Organization/ International Society of Hypertension (WHO/
ISH) risk charts vs Framingham score [1] and WHO/ISH charts, National Cholesterol Education Program - Adult Treatment Panel III (NCEP-ATP III) scores vs Systematic Coronary
Risk Evaluation (SCORE) charts are different [2]. WHO risk charts for the SEAR-B region
developed in 2007 have been validated among Sri Lankans and are the best currently available
risk stratification method for Sri Lankans. The predictions showed 81% agreement between
predictions and observed events but were less predictive in females and those at high CV risk
[3]. Sri Lanka is classified under the SEAR epidemiological sub-region with Indonesia, Cambodia, Laos, Sri Lanka, Maldives, Myanmar, Malaysia, Philippines, Thailand, Timor-Leste,
Viet Nam, Mauritius, and Seychelles [4, 5] but the lifestyle, socio-economic and cultural backgrounds and risk behaviours of Sri Lankans are different to that of people living in other SEAR
countries and therefore may not predict the CVD risk of Sri Lankans exactly. It has been
shown that CVD risk stratification using machine learning of long-term follow-up cohorts is
much more predictive than the risk models developed for a group of countries [6–10].

Therefore, we aimed to develop a CV risk prediction model using long-term follow-up data
from a community-based cohort developed to study metabolic risk factors and non-communicable diseases of Sri Lankans and to validate the new model in an external cohort of Sri Lankans. In addition, we planned to develop an online platform with the best model identified for
practical use of the model in any Sri Lankan.

## Materials and methods

### Study setting

We used 10-year follow-up data of the Ragama Health Study (RHS), a community-based ongoing
study started in 2007 to study the epidemiology of metabolic and non-communicable diseases
(NCD) in Sri Lanka. Details of the Ragama Health Study had been described previously [11].

The baseline cohort in the RHS comprised 35–64-year-old adults resident in the Ragama
Medical Officer of Health (MOH) area (a health administrative area of Sri Lanka). The cohort
was selected randomly and stratified by three age groups (35–44, 45–54 and 55–64 years)
using the voters' list. The study was started on the 1st of February 2007. At baseline in 2007, all
selected individuals were visited at their homes and data on past medical history and behavioural, metabolic and all potential risk factors for NCDs were collected from all consenting

adults. The cohort was re-contacted in 2014 and 2017, and any CVD and new risk factors developed over the follow-up period were documented. All cardiovascular deaths, non-fatal strokes, and non-fatal myocardial infarctions, including elective percutaneous coronary interventions and coronary artery bypass grafts done on patients with symptomatic unstable angina that occurred from 2007 to 2017 were recorded as CVD during these follow-up visits by interviewing patients and their families and perusing clinical notes/death certificates. Of the 2923 participants enrolled at the beginning of the RHS study, 2685 were followed up after 10 years while the remaining 238 could not be traced. The group lost to follow-up was not significantly different from the group followed in terms of gender, age, current smoking status or FBS, SBP, or total cholesterol level at baseline [3]. The study was approved by the Ethics Review Committee of the Faculty of Medicine, University of Kelaniya, Sri Lanka (P38/09/2006). All participants provided written informed consent before enrolment.

## Study population

Of the RHS cohort, we selected participants of 40 years or above who were naïve for CVD at baseline (2007) and were followed up for 10 years up to 2017. Participants who could not be traced in 2017 or whose current status (dead or alive) could not be determined were excluded. Diseased participants whose cause of death could not be verified were excluded from the analysis.

## Data management, model development and statistical analysis

Baseline and follow-up data were extracted from the RHS database. Data extraction and model development started on 08th December 2021. Authors had access to information that could identify individual participants during and after data collection. ML-based models were trained using the database. The missing data in the dataset were strategically addressed through a systematic approach. Initially, null values were filled with the mean, ensuring a representative and unbiased imputation. Subsequently, rows containing null values in crucial features, as identified by Random Forest Classifier (RFC) feature importance, were removed to preserve the integrity of the essential information. The determination of feature importance through RFC contributed to a targeted handling of missing values, prioritising influential features. Duplicates were identified and removed before training ML models to prevent biases, overfitting, and distorted performance metrics and ensured that the models learn from diverse and representative data, to enhance the performance of the model. All potential risk variables were evaluated for inclusion in the model using feature importance analyses conducted with Random Forest and gradient descent algorithms. Synthetic Minority Over-sampling Technique (SMOTE) [12] was applied to overcome class imbalance within the dataset as one class (the individuals without CVD) significantly outnumbered the other class (the individuals with CVD). Using SMOTE we oversampled the minority class, creating synthetic instances, and rectifying the class distribution imbalance in the data set. The oversampling process was parameterized with a specific sampling strategy set at 0.65 or dynamically adjusted if unspecified. 10-fold stratified cross-validation was implemented using the Stratified K-Fold technique [13] to rigorously evaluate the model performance. This method ensured that each cross-validated fold maintains a proportional representation of class instances similar to the original dataset. Within each fold, the dataset was partitioned into a training and a testing dataset. The model was trained on the training data set and evaluated on the testing data set, systematically gauging its robustness and generalization across diverse subsets of data. The combined use of SMOTE for oversampling and Stratified K-Fold for cross-validation provided a comprehensive exploration of the ML-based models' performance characteristics, particularly in the context

of imbalanced data. We developed prediction models using Random Forest Classifier [14]. RFC models were fine-tuned using the Grid Search Cross-Validation approach which involved systematically testing a range of hyperparameter values to identify the combination that produces the best model performance. The performances of the models were assessed through key metrics, including mean F1 score, recall, precision, and accuracy [15]. The mean F1 score was used in selecting the models especially because the data set was unbalanced and it combines precision and recall. A mean F1 score above 0.8 indicates a model with good proficiency in identifying individuals at risk. A precision above 0.75 indicates reliability in the accuracy of positive identifications. A recall score above 0.85 indicates high sensitivity in minimising false negatives. An accuracy exceeding 0.85 reflects a model's overall correctness. Therefore, a model achieving a mean F1 score above 0.8, precision above 0.75, recall above 0.85, and accuracy above 0.85 was considered an effective model in risk prediction. In addition, we used Receiver Operating Characteristic curves also to measure the validity of predictions. We developed three models using different combinations of data of the cohort. In addition, we calculated CV risk predictions of all individuals with the 2019 WHO CV risk models. We calculated WHO risk (lab-based) with R-package [4, 16, 17]. We compared the predictions of the three models and the WHO risk charts.

Secondly, we externally validated the selected best model out of the three in a separate hospital-based database of consecutive patients, 40–74 years of age admitted to Colombo North Teaching Hospital (a tertiary care hospital in Sri Lanka) from 1$^{st}$ of January 2019 to 1$^{st}$ of August 2020 who did not have a history of CVD and presented with an acute incident CVD (acute myocardial infarction or acute stroke) or a disease other than an acute CVD who had complete data for CVD risk calculation. We used this method to increase the yield of CVDs in the validation sample as our aim was to study the accuracy of the new model in predicting incident CVDs. All patients' predicted CVD risks were calculated with the new models and with 2019 WHO risk charts using the most recent pre-morbid risk factor data available up to one year before developing the incident CVD/ admission to the ward. We compared the predictions of the models with observed events using confusion matrices.

Thirdly, we developed an online platform for easy and practical use of the new model among all Sri Lankans.

All statistical analyses were done using SPSS version 22. Categorical data are reported as percentages. Continuous variables are reported as means with standard deviation (SD) or 95% confidence intervals. The significance level was set at p <0.05.

## Ethics approval

Ethics approval was obtained from the Ethics Review Committee of the Faculty of Medicine, University of Kelaniya, Sri Lanka for the original RHS study (P38/09/2006) and the ML development (P61/09/2020). Written informed consent of participants was obtained for the recruitment and follow-up interviews.

## Results

Data of 2596 participants with complete data of 10-year follow-up were selected to train the ML-based models. The baseline characteristics of the study cohort are shown in Table 1. Of them, 179 developed hard CVD events over the 10-year follow-up.

Over the 10-year follow-up period, 179 hard CVD events were recorded; 57 CV deaths, 97 IHD (83 myocardial infarctions, 8 coronary artery bypass grafts, 6 primary percutaneous coronary interventions) and 25 stokes. Of the CVD events, 66 (36.9%) occurred in females and 113 (63.1%) in males.

Three models were developed using different combinations of variables. All the variables according to feature importance were used in Model 1, i.e., age, sex, systolic blood pressure (SBP), diastolic blood pressure (DBP), glycosylated haemoglobin level (HbA1), the ratio of total cholesterol to high-density lipoprotein level (TC/HDL), body mass index (BMI) and the duration of smoking etc. Considering the need of the model for practical use in the community, two more models were developed using freely available data in primary care. Model 2 uses age, sex, smoking status, systolic blood pressure, history of diabetes, and LDL level and model 3 uses age, sex, smoking status, systolic blood pressure, history of diabetes, and total cholesterol level Which are the same variables used in the WHO CV risk prediction charts.

The performance characteristics of the three new models and the WHO risk charts in terms of predicting individuals at high risk of CVD are shown in Fig 1 and Table 2.

Model 1, 2, 3 and the WHO risk charts predicted 173, 162, 169 and 10 out of 179 observed CVDs and the area under the ROC (AUC) were 0.98, 098, 0.98 and 0.52 respectively. All three new ML-based models developed had mean F1 scores above 0.8, precision above 0.75, recall above 0.85, as well as accuracy above 0.85 indicating effective prediction. The WHO risk charts had a good accuracy of 0.92, but recall, was very low indicating a high chance of missing high-risk individuals.

We named the model-3 "Sri Lanka CVD score (SLCVD score)" which is the most practical to use for screening in primary care even in rural communities as it only needs total cholesterol level as a laboratory investigation in calculating the risk. We validated the prediction of SLCVD score, in an external hospital-based cohort of Sri Lankans. The baseline characteristics of the external validation cohort are given in Table 3.

The external validation cohort consisted of 119 patients with incident CVDs and 239 people naïve for CVD. We compared the predictions and observed events of the SLCVD score and the WHO 2019 score in the external validation cohort (Fig 2). SLCVD score and WHO risk

**Table 1. Baseline characteristics of the model development cohort.**

|  | Male | Female | Total |
|---|---|---|---|
|  | **n = 1162** | **n = 1434** | **n = 2596** |
| **Ethnicity n (%)** |  |  |  |
| Sinhalese | 1118 (96.2) | 1375 (95.9) | 2493 (96.0) |
| Tamil | 15 (1.3) | 27 (1.9) | 42 (1.6) |
| Muslim | 2 (0.2) | 2 (0.1) | 4 (0.2) |
| Burgher | 15 (1.3) | 19 (1.3) | 34 (1.3) |
| Other | 12 (1.0) | 11 (0.8) | 23 (0.9) |
| **Age groups (years), n (%)** |  |  |  |
| 40–49.9 | 360 (30.9) | 456 (31.8) | 816 (31.4) |
| 50–59.9 | 526 (45.3) | 669(46.7) | 1195 (46.0) |
| $\geq$ 60.0 | 276 (23.8) | 309(21.5) | 585 (22.6) |
| **Age (years),** mean (SD) | 53.5 (6.8) | 53.4 (6.6) | 53.4 (6.7) |
| **Current smokers,** n (%) | 319 (27.4) | 0 (0.0) | 319 (12.2) |
| **Diabetes,** n (%) | 165 (14.2) | 249 (17.4) | 414 (15.9) |
| **Hyperlipidaemia,** n (%) | 98 (8.4) | 209 (14.6) | 307 (11.8) |
| **SBP (mmHg),** mean (SD) | 134.1 (21.4) | 136.9 (22.5) | 135.7 (22.0) |
| **Total cholesterol (mg/dL),** mean (SD) | 205.4 (39.1) | 219.6 (42.5) | 213.2 (41.6) |
| **BMI (Kg/m$^{2)}$),** mean (SD) | 23.0 (3.9) | 24.8(4.2) | 24.0 (4.2) |

SBP, systolic blood pressure; BMI, body mass index

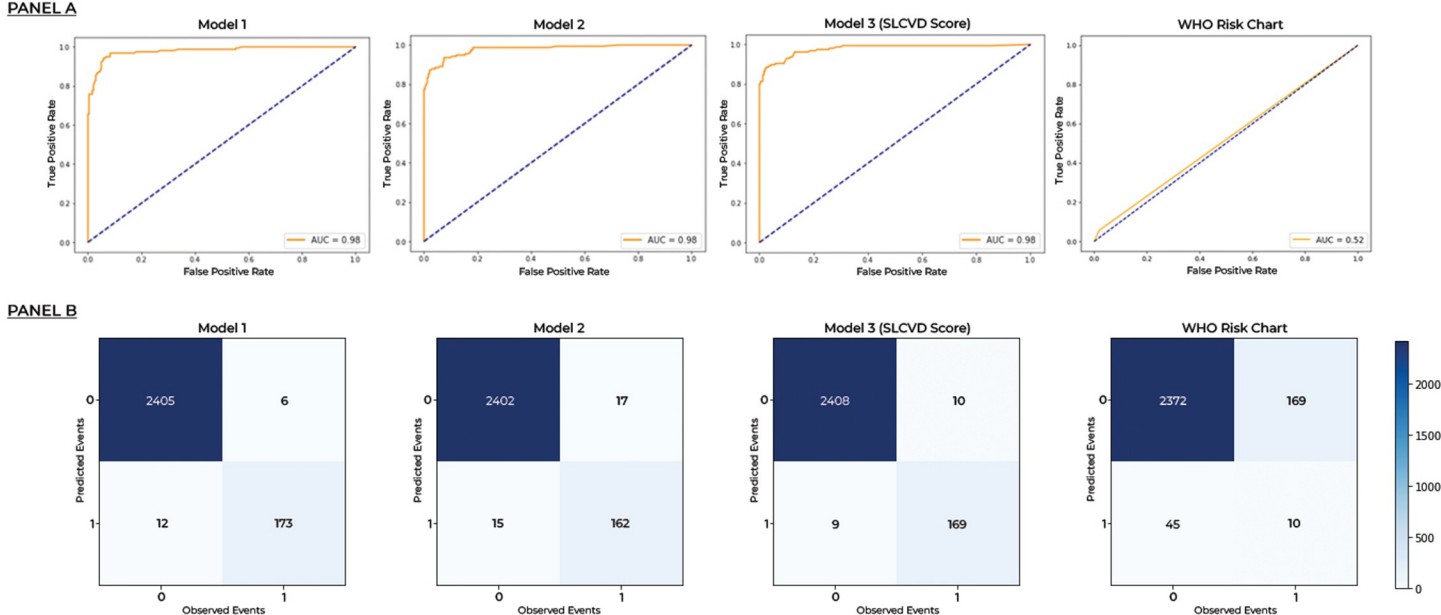

**Fig 1. Comparison of 10-year cardiovascular risk predictions against observed events using different prediction models in the model development cohort.** Panel A–Receiver operating characteristic curves. Panel B–Confusion matrixes.

chart predicted 56 and 18 CV events out of 119 observed events respectively. The SLCVD score was able to predict 38 more cases compared to the predictions of the WHO risk charts. SLCVD score had a 36.1% positive predictive value and 69.0% negative predictive value.

Finally, we developed an online platform to calculate the SLCVD score for easy and practical use of the model among any Sri Lankan anywhere in the country and is available through the link below. SLCVD score calculator.

## Discussion

We developed the first-ever CVD risk prediction model, "SLCVD score", specific for Sri Lankans using machine learning of a Sri Lankan cohort prospectively followed up for 10 years. This is the first CV risk prediction model developed using individual data of Sri Lankans and the only risk prediction model specific to Sri Lankans. The SLCVD score was better in predicting Sri Lankans at high CVD risk than the WHO risk charts developed for the Southeast Asia region. The SLCVD score is simple and uses only a few easily available data in risk stratification enabling its usage even in rural Sri Lanka. We developed an online platform for SLCVD score calculation to make it user-friendly and available for risk stratification of any Sri Lankan anywhere in the world.

The superior ability of ML-based risk assessments than the WHO risk charts in identifying high-risk individuals was observed previously as well in an initial trial of ML model development using all available data variables (75 variables) of the same Sri Lankan cohort [18, 19]. The ability to detect high-risk patients efficiently for primary prevention is very important for a low-middle income country like Sri Lanka, as those are the individuals to benefit mostly by treatment and to be treated aggressively in a risk-based approach than in a blanket population-based approach in primary prevention [20]. Even though the WHO risk charts had good accuracy, it was less sensitive in detecting high-risk individuals. Accuracy is a good measure when the data are quite balanced and when interested in all types of outputs equally [21]. However,

**Table 2. Comparison of the predictive performances of the models.**

|  | Model 1 | Model 2 | Model 3 (SLCVD score) | WHO risk charts |
|---|---|---|---|---|
| **Result** | **High risk** | **High risk** | **High risk** | **High risk** |
| **Precision** | 0.90 | 0.90 | 0.90 | 0.18 |
| **Recall** | 0.92 | 0.93 | 0.93 | 0.06 |
| **F1-Score** | 0.91 | 0.92 | 0.92 | 0.09 |
| **Accuracy** | 0.93 | 0.94 | 0.93 | 0.92 |

as our data set is quite imbalanced and we are more interested in detecting high-risk individuals, accuracy is not the best measure in identifying the best model. It is important not to miss any high-risk individuals as missing a positive case has a much bigger cost than wrongly classifying somebody as having a high risk of CVD. Therefore, maximizing precision and recall is important, and the SLCVD score fared well in the WHO risk charts. Neither precision nor recall is necessarily useful alone since we are interested in the overall picture. F1-score combines precision and recall and works also for cases where the datasets are imbalanced and the SLCVD score had a better F1 score than the WHO risk charts.

ML has been shown to be more effective in predictive events as it carefully studies the behaviours of cohorts over long periods [6–9, 22]. However, using optimal algorithms in developing the ML models is essential [23, 24]. Since the SLCVD score was developed by studying individual data of a cohort of Sri Lankans followed up for 10 years, its likelihood of being more predictive for Sri Lankans than a model developed for an epidemiological region covering several nations without studying individual follow-up data is understandable. Moreover, the satisfactory validation of the SLCVD score in the external validation cohort and the superiority in predicting high-risk individuals of the external validation cohort compared to WHO risk charts attest to the reliability of the predictions of the SLCVD score.

ML-based models using more variables, like model 1, was able to predict more events than the SLCVD score, but we selected the SLCVD score for the online risk calculator development as a simple score using only a few freely available data is the fundamental need of a screening test, especially in a resource-limited setting like Sri Lanka. With the widespread use of smartphones, the online risk calculator can be used by any healthcare or even non-healthcare personnel to calculate CVD risk without having to refer to charts. This will reduce the time spent

**Table 3. Baseline characteristics of the external validation cohort.**

|  | Male | Female | Total |
|---|---|---|---|
|  | **n = 119** | **n = 239** | **n = 358** |
| **Age groups (years),** n (%) |  |  |  |
| 40–49.9 | 6 (5.1) | 12 (5.0) | 18 (5.0) |
| 50–59.9 | 27 (22.9) | 62 (25.8) | 89 (24.9) |
| ≥ 60.0 | 85 (72.0) | 166 (69.2) | 251 (70.1) |
| **Age (years),** mean (SD) | 63.8 (7.7) | 63.2 (6.9) | 63.4 (7.2) |
| **Current smokers,** n (%) | 23 (19.4) | 0 (0.0) | 23 (6.4) |
| **Diabetes,** n (%) | 62 (52.5) | 126 (52.5) | 188 (52.5) |
| **SBP (mmHg),** mean (SD) | 132.9 (17.4) | 130.6 (18.3) | 131.3 (18.0) |
| **Total cholesterol (mg/dL),** mean (SD) | 184.0 (36.9) | 188.0 (33.4) | 186.6 (34.7) |
| **BMI (Kg/m$^{2)}$),** mean (SD) | 24.4 (4.2) | 24.7 (4.1) | 24.6 (4.2) |

SBP, systolic blood pressure; BMI, body mass index

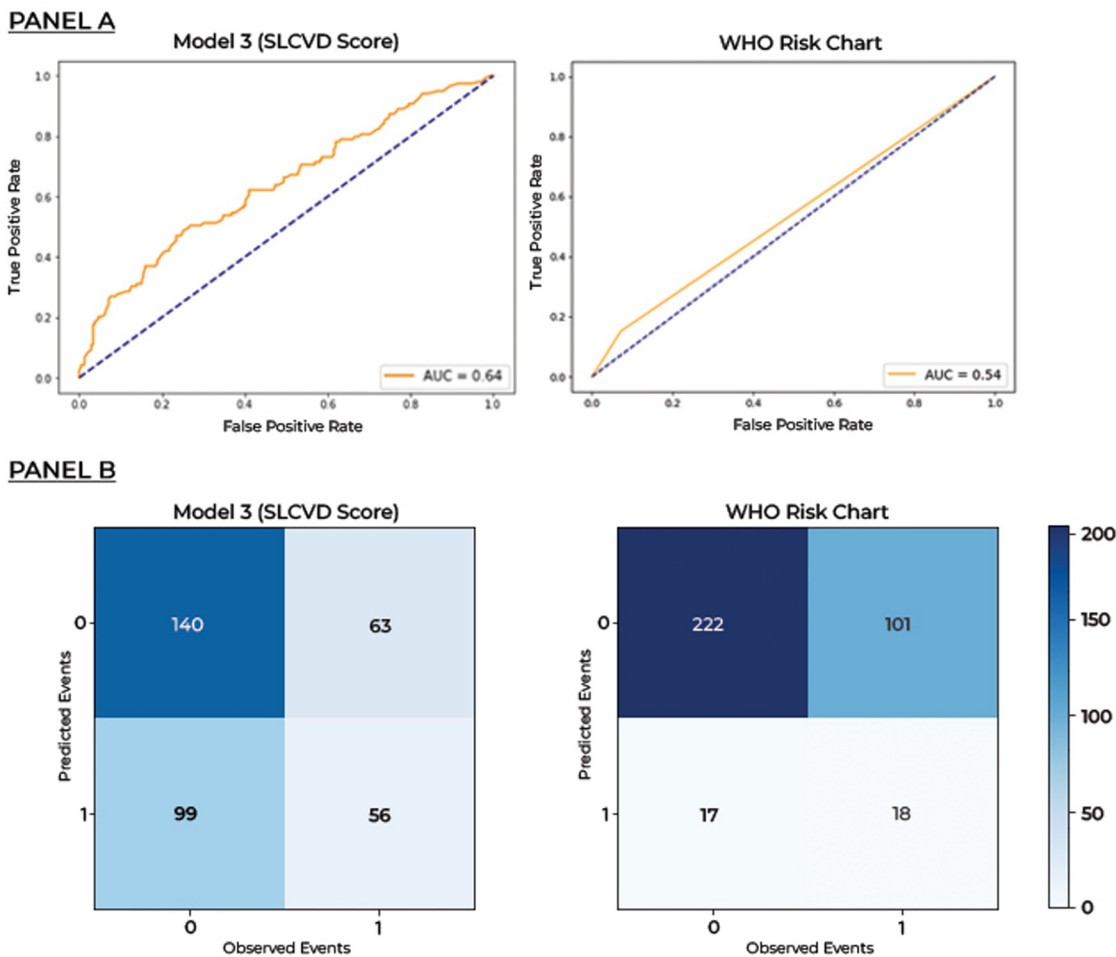

**Fig 2. Comparison of cardiovascular risk predictions against observed events using different prediction models in the external validation cohort.** Panel A–Receiver operating characteristic curves. Panel B–Confusion matrixes.

on risk stratification of individuals and will increase uptake of risk stratification practice in busy primary health and eventually help prevention of non-communicable diseases.

There are several strengths in our study leading to the validity of the SLCVD score. We used data of a Sri Lankan, community-based, randomly selected, semi-urban cohort, individually and prospectively followed up for 10 years to develop the model. Only the data of participants who completed 10-year follow-up were used in the development of the ML model. Patients were followed up by medical officers using face-to-face interviews and medical records and/or death certificates; therefore, self-reporting bias was minimal, and data quality was guaranteed. Hard CVD endpoints were used, and therefore, the data were clear and accurate. We validated the model internally and externally in a separate cohort, and both showed similar predictive performances. We also compared the predictions of the SLCVD score with that of the reference WHO model, and it also showed superiority in predicting high-risk individuals. However, there is one limitation of our study. Even though the cohort we used to train the ML model was a community-based, multi-ethnic random cohort, representation of the estate sector was less in our cohort compared to national distribution. The national distribution of the population in Sri Lanka of urban: rural; estate sectors at the 2012 census was 18.2: 77.4: 4.4 [25] and the same in the Gampaha district where the cohort was drawn from was 15.6: 84.3:

0.1 [26]. However, the percentage distribution of the estate sector is very small compared to the national distribution, and therefore, the effect of this limitation is expected to be minimal. Further, we compared risk factors between cases and controls to develop the model. However, some behavioural risk associations like compliance with medications etc. were not studied. However, we believe that the impact of those factors was minimal as the sample was randomly selected.

In conclusion, we developed the SLCVD score using ML of a Sri Lankan cohort followed up for 10 years for CV risk prediction of Sri Lankans. SLCVD score is more efficient in predicting Sri Lankans at high CVD risk compared to the currently used WHO risk charts developed for the Southeast Asia region. SLCVD score can be calculated using an online calculator which is free to be used by anybody. The score calculator uses only six freely available variables namely, age, sex, smoking status, diabetes status, systolic blood pressure and total cholesterol level and predicts a 10-year risk of developing a hard CVD (i.e.: cardiovascular deaths, non-fatal strokes, and non-fatal myocardial infarctions including elective percutaneous coronary interventions and coronary artery bypass grafts). This calculator is a screening tool valid for any Sri Lankan above 40 years of age who has not had any CVD at the time of calculation. A risk score equal to or more than 20% indicates an individual is at high risk of developing a hard CVD in the next 10 years and they should be aggressively treated with primary preventive measures. The risk scores can be used in executing guideline-based management of hyperlipidaemia and hypertension among Sri Lantanas. Serial risk calculations could be used to objectively study the success of primary prevention interventions.

## Acknowledgments

We thank all who continuously supported the Ragama Health Study, and especially the study participants for their continued cooperation.

**Patient and public involvement statement**

It was not appropriate or possible to involve patients or the public in the design or reporting plans of our research, but it was involved in the conduct and dissemination of the original RHS study. The results of the current study will be disseminated to study participants, other patients and the public following the publication of the study.

## Author Contributions

**Conceptualization:** Chamila Mettananda, Udaya Ranawaka, Arunasalam Pathmeswaran, Anuradhini Kasturiratne, Nei Kato, Rajitha Wickramasinghe, Janaka de Silva.

**Data curation:** Chamila Mettananda, Maheeka Solangaarachchige, Achila Ranasinghe, Isuru Elpitiya, Rashmi Sammandapperuma, Sujeewani Kurukulasooriya, Udaya Ranawaka, Arunasalam Pathmeswaran, Anuradhini Kasturiratne, Nei Kato, Rajitha Wickramasinghe.

**Formal analysis:** Chamila Mettananda, Isuru Sanjeewa, Tinul Benthota Arachchi, Avishka Wijesooriya, Chiranjaya Chandrasena, Tolani Weerasinghe, Maheeka Solangaarachchige, Arunasalam Pathmeswaran, Nei Kato, Prasanna Haddela.

**Funding acquisition:** Chamila Mettananda.

**Investigation:** Chamila Mettananda, Isuru Sanjeewa, Tinul Benthota Arachchi, Avishka Wijesooriya, Chiranjaya Chandrasena, Tolani Weerasinghe, Maheeka Solangaarachchige, Achila Ranasinghe, Sujeewani Kurukulasooriya, Udaya Ranawaka, Arunasalam Pathmeswaran, Nei Kato, Rajitha Wickramasinghe, Prasanna Haddela, Janaka de Silva.

**Methodology:** Chamila Mettananda, Isuru Sanjeewa, Tinul Benthota Arachchi, Avishka Wijesooriya, Chiranjaya Chandrasena, Tolani Weerasinghe, Maheeka Solangaarachchige, Achila Ranasinghe, Isuru Elpitiya, Rashmi Sammandapperuma, Udaya Ranawaka, Arunasalam Pathmeswaran, Anuradhini Kasturiratne, Nei Kato, Rajitha Wickramasinghe, Prasanna Haddela, Janaka de Silva.

**Project administration:** Chamila Mettananda, Achila Ranasinghe, Isuru Elpitiya, Rashmi Sammandapperuma, Sujeewani Kurukulasooriya, Udaya Ranawaka, Anuradhini Kasturiratne, Rajitha Wickramasinghe, Prasanna Haddela.

**Resources:** Chamila Mettananda, Isuru Sanjeewa, Tinul Benthota Arachchi, Avishka Wijesooriya, Chiranjaya Chandrasena, Tolani Weerasinghe, Maheeka Solangaarachchige, Achila Ranasinghe, Isuru Elpitiya, Rashmi Sammandapperuma, Sujeewani Kurukulasooriya, Anuradhini Kasturiratne, Nei Kato, Rajitha Wickramasinghe, Prasanna Haddela, Janaka de Silva.

**Software:** Chamila Mettananda, Isuru Sanjeewa, Tinul Benthota Arachchi, Avishka Wijesooriya, Chiranjaya Chandrasena, Tolani Weerasinghe, Maheeka Solangaarachchige, Nei Kato, Rajitha Wickramasinghe, Prasanna Haddela.

**Supervision:** Chamila Mettananda, Sujeewani Kurukulasooriya, Udaya Ranawaka, Arunasalam Pathmeswaran, Anuradhini Kasturiratne, Prasanna Haddela.

**Validation:** Chamila Mettananda, Isuru Sanjeewa, Tinul Benthota Arachchi, Avishka Wijesooriya, Chiranjaya Chandrasena, Tolani Weerasinghe, Maheeka Solangaarachchige, Arunasalam Pathmeswaran, Prasanna Haddela.

**Visualization:** Chamila Mettananda, Isuru Sanjeewa, Tinul Benthota Arachchi, Avishka Wijesooriya, Chiranjaya Chandrasena, Tolani Weerasinghe, Maheeka Solangaarachchige, Arunasalam Pathmeswaran.

**Writing – original draft:** Chamila Mettananda, Maheeka Solangaarachchige.

**Writing – review & editing:** Chamila Mettananda, Isuru Sanjeewa, Tinul Benthota Arachchi, Avishka Wijesooriya, Chiranjaya Chandrasena, Tolani Weerasinghe, Maheeka Solangaarachchige, Achila Ranasinghe, Isuru Elpitiya, Rashmi Sammandapperuma, Sujeewani Kurukulasooriya, Udaya Ranawaka, Arunasalam Pathmeswaran, Anuradhini Kasturiratne, Nei Kato, Rajitha Wickramasinghe, Prasanna Haddela, Janaka de Silva.

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
