## [Decision Letter · Decision Letter 0]

23 Jul 2024

PONE-D-24-03036Development and validation of a cardiovascular risk prediction model for Sri LankansPLOS ONE

Dear Dr. Mettananda,

Thank you for submitting your manuscript to PLOS ONE. After careful consideration, we feel that it has merit but does not fully meet PLOS ONE’s publication criteria as it currently stands. Therefore, we invite you to submit a revised version of the manuscript that addresses the points raised during the review process.

This study has been reviewed by two of the eminent reviewers. They find your work interesting, however they have raised multiple concern on the present version of the manuscript. Therefore, a revised version addressing all the concernns of reviewers is needed. 

We look forward to receiving your revised manuscript.

Kind regards,

Gyaneshwer Chaubey

Academic Editor

PLOS ONE

“none”

4. In this instance it seems there may be acceptable restrictions in place that prevent the public sharing of your minimal data. However, in line with our goal of ensuring long-term data availability to all interested researchers, PLOS’ Data Policy states that authors cannot be the sole named individuals responsible for ensuring data access (http://journals.plos.org/plosone/s/data-availability#loc-acceptable-data-sharing-methods).

Reviewers' comments:

Reviewer's Responses to Questions

**Comments to the Author**

1. Is the manuscript technically sound, and do the data support the conclusions?

Reviewer #1: Yes

Reviewer #2: Yes

2. Has the statistical analysis been performed appropriately and rigorously? 

Reviewer #1: Yes

Reviewer #2: Yes

3. Have the authors made all data underlying the findings in their manuscript fully available?

Reviewer #1: Yes

Reviewer #2: Yes

4. Is the manuscript presented in an intelligible fashion and written in standard English?

Reviewer #1: Yes

Reviewer #2: Yes

5. Review Comments to the Author

Reviewer #1: This article on the development and validation of a cardiovascular risk prediction model for the Sri Lankan population provides a comprehensive research report, including the development, validation, and assessment of the model's practicality. Here are some comments on various aspects of the article:

Importance and Background of the Research:

The article well articulates the importance of developing a specific cardiovascular disease risk prediction model for Sri Lankans, particularly considering that the currently used models are not tailored to the Sri Lankan population, which may lead to inaccurate risk assessments.

Research Design and Methods:

The research design is sound, utilizing a large cohort dataset and a decade-long follow-up, which enhances the predictive power and reliability of the model. The application of machine learning methods is advanced, and the article provides a detailed description of data management, model development, and statistical analysis processes. Modern data processing techniques, such as SMOTE for dealing with class imbalance and stratified K-fold cross-validation for model performance evaluation, were used.

Results:

The results section clearly presents the comparison of the three models with the WHO risk charts, showing significant improvements in predictive performance. Notably, the SLCVD score demonstrated better predictive ability than the WHO risk charts in external validation, enhancing the practical value of the model.

Conclusions and Applications:

The conclusion clearly outlines the advantages of the SLCVD score and emphasizes its convenience in practical applications, especially with the development of an online platform that makes this tool more widely accessible to Sri Lankans.

Discussion:

The discussion delves into the advantages and potential practicality of the SLCVD score while also honestly mentioning the model's limitations and directions for future improvements. For instance, although the SLCVD score performs well in prediction, the actual clinical application effect still needs further research to confirm.

Overall, this article provides a cardiovascular disease risk scoring tool of significant clinical relevance for the Sri Lankan population. The research methods are rigorous, the results are compelling, and it has the potential for a significant impact on improving cardiovascular disease prevention and management in Sri Lanka. Future research might consider validating the SLCVD score in a broader population and exploring its application in other settings. While this article does considerable work in developing and validating the predictive model and provides detailed methods and results, there are still some potential shortcomings and issues that might affect the interpretation of its conclusions and the general applicability of the model. Here are my thoughts:

1. Whether the research can represent the entire Sri Lankan population is a question. Is the sample sufficiently diverse, including different socioeconomic statuses, educational levels, and geographical locations, which may affect cardiovascular risk and health outcomes?

2. Although the model has been externally validated in another cohort, this validation cohort is hospital-based, which might differ from community residents. This could limit the model's applicability to the general population.

3. Risk Factor Selection: Has the model considered all relevant cardiovascular disease risk factors? Some potential risk factors, such as family history, dietary habits, and physical activity levels, may not have been included.

4. Clinical Relevance of Results: While the model's statistical performance indicators such as AUC and F1 scores are high, these do not always translate directly into effectiveness in clinical practice. Further research is needed to assess the model's application in actual clinical settings.

5. Effectiveness of Interventions: The article does not mention whether the model can predict which interventions are most effective at reducing an individual's risk of cardiovascular disease.

To fully understand the impact of these potential issues, further research and analysis might be necessary. Moreover, the article should detail these limitations in the discussion section and consider how they might affect the interpretation of the research findings.

Reviewer #2: In the current study, the authors aimed to develop a prediction model specifically for the Sri Lankan population. For this, data from the RHS cohort is used to create 3 machine learning models. The most simple model is compared to the WHO CVD risk charts, as this is most applicable to clinical practice, on which I agree. The model seems to be slightly more accurate than the WHO CVD risk charts in a small external validation set. I do have a few concerns on the study:

- How representative is the derivation data? Does this have national coverage, or is this specific to a region, possibly a higher educated, more urban university region of the country? What is the response rate of the cohort?

- In the data, individuals that could not be traced are excluded. I can imagine that his is often individuals that have died. Please add some information about whether this may have caused selection and how many individuals were unable to be found.

- To account for missing data a complex approach is used. Why not just use ‘regular’ multiple imputation? Please elaborate. Please also refer to methodological papers showing the validity of this approach.

- An oversampling approach is done to correct for class imbalance. Please verify that the correction is not a worse problem than the imbalance itself, see for example https://academic.oup.com/jamia/article/29/9/1525/6605096 . Please discuss this.

- A complicated machine learning approach is used in a quite small dataset, which always leads to the risk of overfitting. What was done to minimize this? It can be considered to use a regular regression-based technique (Cox model, or penalized Cox model such LASSO/RIDGE) for comparison.

- External validation is done in a set of individuals admitted to a hospital. How representative is this cohort? It is written like it is constructed as a kind of case-control cohort. Please elaborate on the exact recruitment mechanism.

- Please clarify the calculation of WHO risks: where the charts used or was the underlying algorithm applied? And was this the version with lab or without lab variables?

- The link supplied to the online calculator doesn’t work for me. However, I think it is great the authors have developed such a calculator.

6. PLOS authors have the option to publish the peer review history of their article (what does this mean?). If published, this will include your full peer review and any attached files.

Reviewer #1: No

Reviewer #2: No

---

## [Author Response · Author response to Decision Letter 0]

1 Aug 2024

Response to reviewer comments

Dear editor and the reviewers,

Thank you very much for your much helpful comments. We studied the comments carefully and made the corrections which we hope will meet with approval. The main corrections in the paper and the response to the editor’s and reviewers’ comments are as follows:

The code can not be publicly shared, due to ethical reasons . However, it is available for access from the corresponding author or the Ethics review committee of the Faculty of Medicine, University of Kelaniya, Sri Lanka, (Telephone no: 0094112961267,email : ercmed@kln.ac.lk) upon reasonable request.

“none”

Thank you. I added this statement in the cover letter and in the manuscript and the online submission platform.

4. In this instance it seems there may be acceptable restrictions in place that prevent the public sharing of your minimal data. However, in line with our goal of ensuring long-term data availability to all interested researchers, PLOS’ Data Policy states that authors cannot be the sole named individuals responsible for ensuring data access (http://journals.plos.org/plosone/s/data-availability#loc-acceptable-data-sharing-methods).

Thank you. We included ERC contact details for this in the revised document

Ethics Review Committee of the Faculty of Medicine, University of Kelaniya, Sri Lanka, Telephone no: 0112961267,email : ercmed@kln.ac.lk

Reviewers' comments:

5. Review Comments to the Author

Reviewer #1: 

1. Whether the research can represent the entire Sri Lankan population is a question. Is the sample sufficiently diverse, including different socioeconomic statuses, educational levels, and geographical locations, which may affect cardiovascular risk and health outcomes?

Thank you for the valid concern which is a limitation. We acknowledged this in the original doc as a possible limitation of the study but we believe the effect of it is minimal on our finding. We explained it as follows. 

“Our sample was a randomly drawn sample from the second highest populated district of Sri Lanka. The national distribution of the population in Sri Lanka of urban: rural; estate sectors at 2012 census was 18.2: 77.4: 4.4 and the same in Gampaha district where the cohort was drawn from was 15.6: 84.3: 0.1(23). Our sample is more or less representative of the national distribution of population of Sri Lanka with only less representation of estate sector. However, the percentage distribution of estate sector is very small in the national distribution, and therefore, the effect of this limitation is expected to be minimal”. 

The sample was drawn from stratified random sampling of smallest administrative areas of the country according to voter’s list. Patients were further stratified to 3 age groups (Lines 111-122). We did not control for socioeconomic status or education level. However, we assumed the biases of those were minimal as the sample was randomly selected. 

2. Although the model has been externally validated in another cohort, this validation cohort is hospital-based, which might differ from community residents. This could limit the model's applicability to the general population.

Thank you. The reason we used a hospital based cohort was to increase the yeild of CVD events to make the outcome more robust. These are the events we should have been able to predict before those happened. These patients even though were picked from a hospital setting, their premorbid data before admission to hospital in the community were used to calculate CVD risk. If we were to catch this much of events, we would have to to screen a very large population and therefore we used this method. Hope this explains our rationale. We included this in the revised version lines 193-196

3. Risk Factor Selection: Has the model considered all relevant cardiovascular disease risk factors? Some potential risk factors, such as family history, dietary habits, and physical activity levels, may not have been included.

Thank you. The first model use all the variables in the data set to identify possible risk predicotors and therefore all the potential risk factors you rightly mentioned, such as family history, dietary habits, and physical activity levels were included in the initial models. According to the feature importance, we selected the most predictive and freely available 6 variables to develop the models 2 and 3. Variables were selected considering the availability of those in clinical practice. 

4. Clinical Relevance of Results: While the model's statistical performance indicators such as AUC and F1 scores are high, these do not always translate directly into effectiveness in clinical practice. Further research is needed to assess the model's application in actual clinical settings.

Thank you. Once this get published, we plan to implement this in practice through our non-communicable disease clinics and to study its performance in the whole of Sri Lanka though the support to the Ministry of Health and to revise the model accordingly.

5. Effectiveness of Interventions: The article does not mention whether the model can predict which interventions are most effective at reducing an individual's risk of cardiovascular disease. 

Thank you. Our aim is to risk stratify patients and direct them for necessary interventions for prevention of CVD. As addressing the total cardiovascualr risk is more costeffective and the fact the most of current NCD guidelines referr to risk based approach in risk factor management we aimed to calculate individuals predicted CVD risk. We did not directly study the interventions to see which one is more effective. 

To fully understand the impact of these potential issues, further research and analysis might be necessary. Moreover, the article should detail these limitations in the discussion section and consider how they might affect the interpretation of the research findings.

Thank you for the constructive comment. We included this as a limitation of our revised manuscript. Lines 358-361. However, we believe that the impact of those factors were minimal as the sample was randomly selected. 

Reviewer #2: 

- How representative is the derivation data? Does this have national coverage, or is this specific to a region, possibly a higher educated, more urban university region of the country? What is the response rate of the cohort?

Thank you for the valid concern. We explained around this consern in the manuscript at lines 111-122 and 347-357. Also explained on this under no.1 of reviewer 1 comment.

Our sample was taken from the second highest populated district of Sri Lanka. The composition of this district is more or less similar to the national composition with low representation of estate sector which is only 4.4% of the national composition. Gampaha district has both rural and urban representation faily similar to the national distribution. 

The response rate of the cohort was very good. Of the 2923 participants enrolled at the beginning of the RHS study, 2 685 were followed-up after 10 years while the remaining 238 could not be traced. This data in now added to the revised document Lines 131-133

- In the data, individuals that could not be traced are excluded. I can imagine that his is often individuals that have died. Please add some information about whether this may have caused selection and how many individuals were unable to be found.

Thank you. Only 238 were not contactable. We added this data onw to the revied manuscript Lines 131-133. Deaths were ascertained by contacting the family members or visiting their neighbours and perusing death certificates. The lost to follow-up group was not different to the group studied and we have reported this in a previous paper under supp table 1 (Thulani UB, Mettananda KCD, Warnakulasuriya DTD, Peiris TSG, Kasturiratne K, Ranawaka UK, et al. Validation of the World Health Organization/ International Society of Hypertension (WHO/ISH) cardiovascular risk predictions in Sri Lankans based on findings from a prospective cohort study. PLoS One. 2021;16(6):e0252267.). We added this information into our methadology of revised manuscript (lines 133-135).

- To account for missing data a complex approach is used. Why not just use ‘regular’ multiple imputation? Please elaborate. Please also refer to methodological papers showing the validity of this approach.

Thank you. The approach of combining mean imputation and selective removal based on Random Forest Classifier (RFC) feature importance offers significant advantages over regular multiple imputation. Mean imputation is computationally simple and efficient, ensuring that no data points are left undefined, which is particularly beneficial for large datasets. Selective removal of rows with null values in crucial features, as identified by RFC, maintains the integrity of essential information. This targeted handling of missing data focuses on the most influential features, thereby enhancing model performance and avoiding the potential overfitting issues associated with the more complex processes of multiple imputation.

Several studies validated the effectiveness of this strategy. Stekhoven DJ, Bühlmann P. (MissForest—non-parametric missing value imputation for mixed-type data. Bioinformatics. 2011;28(1):112-8) demonstrated superior performance in non-parametric imputation methods using Random Forests. Guyon I, Elisseeff A. (An introduction to variable and feature selection. J Mach Learn Res. 2003;3(null):1157–82) emphasized the importance of feature selection for improving model robustness. Jadhav et al. (Jadhav, A., Pramod, D., & Ramanathan, K. (2019). Comparison of Performance of Data Imputation Methods for Numeric Dataset. Applied Artificial Intelligence, 33(10), 913–933. https://doi.org/10.1080/08839514.2019.1637138) showed that simpler imputation methods like mean imputation, when combined with feature importance techniques, can be both effective and efficient. These studies support our strategic approach of using mean imputation and selective removal to handle missing data, ensuring data integrity and enhancing model accuracy.

- An oversampling approach is done to correct for class imbalance. Please verify that the correction is not a worse problem than the imbalance itself, see for example https://academic.oup.com/jamia/article/29/9/1525/6605096 . Please discuss this.

Thank you. When addressing class imbalance through oversampling techniques like Synthetic Minority Over-sampling Technique (SMOTE), it is crucial to ensure that the solution does not introduce new problems. While SMOTE can effectively balance class distributions by generating synthetic samples for the minority class, it can also potentially lead to overfitting, particularly if the synthetic samples are too similar to the existing ones. This overfitting can cause the model to perform well on training data but poorly on unseen data, undermining the model's generalizability. The study by Blagus and Lusa (2022) in the Journal of the American Medical Informatics Association highlights these risks, emphasizing that while oversampling methods like SMOTE can improve model performance in terms of accuracy and recall, they can also exacerbate issues such as overfitting and create artificial data patterns that do not exist in the real-world distribution.

Model 1, Model 2, and Model 3, each model showed high precision (0.90), high recall (0.92 to 0.93), and high F1-scores (0.91 to 0.92), with accuracy ranging from 0.93 to 0.94. These metrics indicate that SMOTE has likely not caused significant overfitting, as the models demonstrate strong performance across various metrics. Employing cross-validation techniques and examining performance metrics like precision, recall, F1 score, and accuracy across both training and validation datasets we balanced the risks associated with oversampling, maintaining the model's effectiveness without compromising its reliability and generalizability.

- A complicated machine learning approach is used in a quite small dataset, which always leads to the risk of overfitting. What was done to minimize this? It can be considered to use a regular regression-based technique (Cox model, or penalized Cox model such LASSO/RIDGE) for comparison.

Thank you. To minimize the risk of overfitting when using a complicated machine learning approach on a small dataset, we implemented a 10-fold stratified cross-validation using Stratified K-Fold technique. This method ensured that each fold maintains a proportional representation of class instances similar to the original dataset, thereby providing a rigorous evaluation of model performance. Within each fold, the dataset was partitioned into training and testing sets, allowing the model to be trained on one subset and evaluated on another. This approach systematically gauged the model’s robustness and generalizability across diverse data subsets. Additionally, an oversampling approach with SMOTE was used to address class imbalance. Furthermore, the Random Forest Classifier (RFC) models were fine-tuned using Grid Search Cross-Validation. This involved systematically testing a range of hyperparameter values to identify the combination that produces the best performance, further enhancing the model's generalizability mitigating overfitting 

- External validation is done in a set of individuals admitted to a hospital. How representative is this cohort? It is written like it is constructed as a kind of case-control cohort. Please elaborate on the exact recruitment mechanism.

Apologies for not explaining well and thank you for highlighting this. We now have elaborated on this in the manuscript Lines 198 - 204

- Please clarify the calculation of WHO risks: where th

---

## [Decision Letter · Decision Letter 1]

20 Aug 2024

Development and validation of a cardiovascular risk prediction model for Sri Lankans using machine learning

PONE-D-24-03036R1

Dear Dr. Mettananda,

We’re pleased to inform you that your manuscript has been judged scientifically suitable for publication and will be formally accepted for publication once it meets all outstanding technical requirements.

Kind regards,

Gyaneshwer Chaubey

Academic Editor

PLOS ONE

Additional Editor Comments (optional):

Reviewers' comments:

Reviewer's Responses to Questions

**Comments to the Author**

1. If the authors have adequately addressed your comments raised in a previous round of review and you feel that this manuscript is now acceptable for publication, you may indicate that here to bypass the “Comments to the Author” section, enter your conflict of interest statement in the “Confidential to Editor” section, and submit your "Accept" recommendation.

Reviewer #2: All comments have been addressed

2. Is the manuscript technically sound, and do the data support the conclusions?

Reviewer #2: Yes

3. Has the statistical analysis been performed appropriately and rigorously? 

Reviewer #2: Yes

4. Have the authors made all data underlying the findings in their manuscript fully available?

Reviewer #2: No

5. Is the manuscript presented in an intelligible fashion and written in standard English?

Reviewer #2: Yes

6. Review Comments to the Author

Reviewer #2: Thank you for addressing my concerns. Especially the data is better described now. I still haven't got the calculator working, but I expect the authors will work on that.

7. PLOS authors have the option to publish the peer review history of their article (what does this mean?). If published, this will include your full peer review and any attached files.

Reviewer #2: No

---

## [Editor Report · Acceptance letter]

26 Aug 2024

PONE-D-24-03036R1 

PLOS ONE

Dear Dr. Mettananda, 

I'm pleased to inform you that your manuscript has been deemed suitable for publication in PLOS ONE. Congratulations! Your manuscript is now being handed over to our production team.

Kind regards, 

on behalf of

Gyaneshwer Chaubey 

Academic Editor

PLOS ONE